# Expression of Neurotrophins and Its Receptors During Fetal Development in the Human Cochlea

**DOI:** 10.3390/ijms252313007

**Published:** 2024-12-03

**Authors:** Claudia Steinacher, Shin-ya Nishio, Shin-ichi Usami, Jozsef Dudas, Dietmar Rieder, Helge Rask-Andersen, Berta Crespo, Nadjeda Moreno, Marko Konschake, Christof Seifarth, Rudolf Glueckert

**Affiliations:** 1Department of Otorhinolaryngology, Medical University Innsbruck, 6020 Innsbruck, Austria; claudia.steinacher@i-med.ac.at; 2Department of Hearing Implant Sciences, Shinshu University School of Medicine, Matsumoto 3-1-1 Asahi, Nagano 390-8621, Japan; nishio@shinshu-u.ac.jp (S.-y.N.); usami@shinshu-u.ac.jp (S.-i.U.); 3Institute of Bioinformatics, Medical University Innsbruck, 6020 Innsbruck, Austria; dietmar.rieder@i-med.ac.at; 4Department of Surgical Sciences, Otorhinolaryngology and Head and Neck Surgery, Uppsala University, 751 85 Uppsala, Sweden; helge.rask-andersen@uu.se; 5UCL Great Ormond Street Institute of Child Health, University College London, London WC1 N1EH, UK; berta.crespo@ucl.ac.uk (B.C.); n.moreno@ucl.ac.uk (N.M.); 6Institute of Clinical and Functional Anatomy, Medical University Innsbruck, 6020 Innsbruck, Austria; marko.konschake@i-med.ac.at (M.K.); christof.seifarth@i-med.ac.at (C.S.)

**Keywords:** human inner ear, BDNF, NT-3, NT-4, Trk receptor, p75, neurotrophins, development, hearing

## Abstract

We determined the relative expression levels of the receptors *TrkA*, *TrkB*, *TrkC*, and *p75^NTR^* and ligands *BDNF*, *NT-3*, *NGF*, and *NT-4* with RNAseq analysis on fetal human inner ear samples, located TrkB and TrkC proteins, and quantified *BDNF* with in situ hybridization on histological sections between gestational weeks (GW) 9 to 19. Spiral ganglion neurons (SGNs) and satellite glia appear to be the main source of *BDNF* and synthesis peaks twice at GW10 and GW15–GW17. Tonotopical gradients of *BDNF* revert between GW8 and GW15 and follow a maturation and innervation density gradient in SGNs. *NT-3/TrkC* follows the same time course of expression as *BDNF/TrkB*. Immunostaining reveals that TrkB signaling may act mainly through satellite glia, Schwann cells, and supporting cells of Kölliker’s organ, while TrkC signaling targets SGNs and pillar cells in humans. The *NT-4* expression is upregulated when *BDNF/NT-3* is downregulated, suggesting a balancing effect for sustained TrkB activation during fetal development. The mission of neurotrophins expects nerve fiber guidance, innervation, maturation, and trophic effects. The data shall serve to provide a better understanding of neurotrophic regulation and action in human development and to assess the transferability of neurotrophic regenerative therapy from animal models.

## 1. Introduction

The inner ear is a complex organ with hard bone and vast fluid spaces, and contains some of the most specialized tissue within the mammalian body. A precisely orchestrated coordination of the migration and differentiation of cells builds the complex architecture of the labyrinth. The genetic and molecular aspects of these processes are more and more uncovered and the involvement of genes highly related to hearing loss are in focus in order to evolve new therapeutic strategies. Future hair cell and auditory neuron regeneration therapies may recapitulate some of the developmental processes like cellular pattering, hair cell innervation, and neuron wiring. This necessitates a deeper understanding of the main signaling pathways of sensorineural development in humans and decipher the functional implications [1,2,3,4].

The human inner ear starts to form out of the otic pit invagination (around day 23–26/Carnegie stage 11). Subsequently, the otic vesicle forms from ectodermal cells around gestational week (GW) 04. Between GW04 and GW05, the vestibular pouch starts to form the semicircular canals, followed by the utricle and saccule. The cochlea starts to sprout from the cochlear pouch around GW08 and reaches the 2.5 coiled turns around GW10 [5,6,7,8]. The cochlear duct is a tube that contains the sensory epithelium and the fluid compartment of the scala media with its unique ion composition essential for later hearing function. It is the only cochlear fluid compartment present at early stages and elongates in a spiral way. The greater epithelial ridge (GER) and lesser epithelial ridge (LER) are the cell-rich parts of the cochlear duct and later form the complex sensory apparatus of the organ of Corti. As described by Kelley, M. [9], the GER contains the cell-rich bulge of Kölliker’s organ, inner hair cells (IHCs), and the supporting cells, the inner phalangeal and inner pillar cells. Similarly, the LER contains the outer hair cells (OHCs), outer pillar cells and cells of the future outer sulcus [8,9,10]. IHCs are the primary receptor cells that turn vibrations into auditory information, while OHCs serve to actively amplify and sharpen the vibration of the basilar membrane.

The cell cycle exit of hair cells in the human cochlea starts as early as week 7 in the apex [5]. Around GW11–12, the differentiation of IHCs starts with an opposing gradient in a basal-to-apical sequence. Two weeks later, the differentiation of OHCs begins in the middle turn region [6,8,11]. Both the hair cell differentiation and innervation of hair cells by spiral ganglion neurons (SGNs) are critical steps in development. At GW08, SGNs are located adjacent to the GER and around GW10, and the neuronal tissue “moves” centrally to form the future central spindle termed modiolus, with SGNs comprising in a spiral canal. Simultaneously to the centering of SGNs, the peripheral nerve fibers start to elongate and innervate the hair cells [5,6,8,12,13].

Neurotrophins play a pivotal role in these developmental processes and regulate neural survival and axon guidance [14]. There are four known neurotrophins in mammals: nerve growth factor (NGF), brain-derived neurotrophic factor (BDNF), neurotrophin-3-(NT-3), and neurotrophin-4 (NT-4). Neurotrophins are synthesized and expressed in end-organs like hair follicles, in Schwann cells, and in fibroblasts after the release of cytokines from macrophages that infiltrate nerve tissue as parts of an inflammatory response. Mast cells produce neurotrophins upon activation and in damaged nerves to promote regeneration and enhance the survival of injured neurons [15,16,17,18].

The most prevalent neurotrophic proteins in the inner ear are BDNF and NT-3, both of which bind to distinct receptors with a different affinity to trigger a response. Neurotrophins act as a ligand at tropomyosin-related kinase receptors (Trk), which include the high-affinity TrkA (NTRK1), TrkB (NTRK2), TrkC (NTRK3), and the low-affinity p75 neurotrophin receptor (p75^NTR^) [15,17,18,19]. Expression profiling in null mutant mice models showed a trophic role of BDNF and NT-3 on sensory neurons in the inner ear [20]. Previous studies on human fetal ganglions have shown that the expression of neurotrophins and their receptors start with TrkA at GW05. The expression of TrkA vanishes after GW09 at the protein level [21]. TrkB and TrkC reached a peak expression between GW08 and GW12, while p75^NTR^ is more or less constantly expressed throughout inner ear development [20,21]. The exact role of p75^NTR^ in the development of the inner ear and in adults is still unclear but is supposed to increase the binding affinity to Trk receptors, thereby boosting Trk signaling [22]. This pattern of receptor expression coincides with specific developmental events like the innervation of hair cells as well as bigger morphological changes [21]. A lack in the synthesis or secretion of neurotrophins can result in a tonotopical distinct reduction in SGNs or a complete degeneration during adulthood [23,24]. Mapping the expression of neurotrophins and their receptors during human development is important in order to understand the role of neurotrophins in inner ear tissue formation [25,26,27]. We previously determined the expression patterns of BDNF, p75^NTR^, and TrkB&C receptors in the developing human fetal inner ear between GW09 to GW12 [21], and now extend our analysis from GW09 to GW19 to also cover the later fetal developmental stages and add RNAseq data from that period.

Mapping the spatiotemporal expression of neurotrophins and their receptors is essential for understanding their roles in inner ear development. Previously, we characterized the expression of BDNF, p75NTR, and TrkB/C receptors in the human fetal inner ear from GW9 to GW12. In this study, we extend the analysis to GW19, incorporating RNAseq data, to capture the later stages of fetal development.

## 2. Results

For the analysis of BDNF, TrkB&C, and p75^NTR^ expression during the development of the human inner ear, we focused on the sensory epithelia and spiral ganglion. We used fetal inner ears from early GW09, where the cochlear duct is the only fluid space in the cochlea, to a later time point (GW19) that resembles almost a functional mature hearing organ.

### 2.1. Expression of Neurotrophins and Neurotrophic Receptors with RNAseq Analysis

To determine the gene expression level changes of *BDNF*, we used RNAseq analyses of cochlear tissue without the vestibular system in order to track the profiles of up- and downregulation across fetal development. Relative expression levels were calculated using gestational week 11 as a reference. Facial nerve tissue is associated so closely with the cochlea that some part of the VII^th^ cranial nerve may adhere in our preparations. The relative expression of the receptors *TrkB*, *TrkC*, and *p75^NTR^* in relation to GW11 show a very similar course with the peak expression at GW15/GW16. *TrkC* remains upregulated until GW18. *TrkA* transcripts are more downregulated than other Trk receptors compared to GW11. Since we found the protein previously only in the facial nerve at the early gestational stages (GW10) [21], we regard the *TrkA* levels as not relevant for fetal cochlear development. The ligands *BDNF* and *NGF* peak at GW16 and *NT-3* a week earlier at GW15. *NT-4* that also binds to the *TrkB* receptor showed an opposing biphasic expression from GW12-14 and was upregulated at GW18-19 (Figure 1). Classical heat map visualization is added as a Appendix A.

### 2.2. Expression Profile of BDNF in the Cochlea with In Situ Hybridization

Due to the limited availability of human fetal material for the histology, we had to focus our ISH on the detection of *BDNF*. *BDNF* transcripts are distributed at GW10 not only across sensorineural structures. Staining is present in mesenchymal tissue that later forms the perilymphatic spaces and in the bone of the otic capsule (Figure 2A). At GW12, ISH showed a drastic reduction in expression in the bone and rather weak staining in the cochlear duct and spiral ganglion (Figure 2C). This is in line with the RNAseq data profiles and marks the formation of scala vestibuli and, additionally, scala tympani in the basal turn. GW15 is close to the *BDNF* peak expression in RNAseq data and reveals an upregulation and further concentration of the ISH signal in the SGNs and sensory epithelium. A weaker staining is present in the lateral wall adjacent to the stria vascularis (Figure 2E). All fluid compartments are present and the modiolus formed. *BDNF* faded at GW18 (Figure 2G), which confirms the trend in whole-transcriptome sequencing. For each specimen, we used consecutive sections for the sense ISH and antisense ISH. None of the sections with a sense probe yielded any positive reaction (Figure 2B,D,F,H).

The results from both gene expression methods (RNAseq and ISH) of the *BDNF* largely overlap, which is an indicator for the reliable detection of *BDNF* expression changes in the human cochlea. We further focused our analysis on the sensory epithelium and spiral ganglion neurons to quantify and better locate the expression at the subcellular level.

### 2.3. Expression of BDNF, TrkB, TrkC, and p75^NTR^ in the Spiral Ganglion

#### 2.3.1. BDNF RNA in the Spiral Ganglion

We measured *BDNF* transcripts along the tonotopical axis of SGNs to account for any gradients from the base to apex. The *BDNF* ISH hybridization results are plotted as the sum intensity of staining and correspond to the total RNA transcripts within a region of interest (ROI). The results (Figure 3A) showed that the staining intensity in the spiral ganglion peaked at GW10 and exposed a clear tonotopical gradient with the highest staining intensity in the basal turn. By GW12, BDNF production is downregulated until GW14. At GW15, we observed a second peak of the expression of *BDNF* in the middle turn, followed by a gradual decrease at the later stages. In the apical turn, the second peak appears two weeks later at GW17, followed by a drastic drop. Interestingly, relative *BDNF* levels in the middle turn fluctuated at considerably lower levels of staining intensity within this period. From GW18 to GW19, the *BDNF* transcripts are largely shut down.

In detail, the GW10 sections exhibit the highest staining intensities for *BDNF* in the spiral ganglion that is concentrated as a cell aggregate very close to the cochlear duct and also contains smaller glial cells. Some bigger cells that we ascribe to auditory neurons show the maximum intensity. Apical, middle, and basal turn SGNs produce *BDNF* transcripts (Figure 3B,D,F), as well as satellite glia (Figure 3D inset) and Schwann cells (Figure 3F inset). At GW12, the *BDNF* ISH signal is higher in glial cells around the spiral ganglion body, and only scattered transcripts appear intracellularly in SGNs (Figure 3H). The ISH signal increased between GW15 and GW16 in the middle turn (Figure 3J) and fades in nerve fibers and Schwann cells. Basal turn SGNs are less densely arranged with a lower staining intensity (Figure 3L). From GW18 to GW19, the distribution of the *BDNF* ISH signal remained confined to satellite glia cells and SGNs (Figure 3N). For each developmental stage, we used sequential sections to pair the sense and anti-sense probes for a better comparison (Figure 3C,E,G,I,K,M,O).

A distinct basal–apical *BDNF* gradient is present only at GW10, which becomes more complex at the later developmental stages.

#### 2.3.2. Neurotrophic Receptor Immune Staining

To correlate the spatiotemporal expression pattern of *BDNF* with the activating receptors, we performed immune staining for the TrkB receptor and p75^NTR^ from GW09 to GW19 and additionally added TrkC to account for the possible involvement of NT-3.

By GW10, the positive immunoreactivity (IR) for TrkB and TrkC was limited to the spiral ganglion somata and nerve fibers (Figure 4A,A.1,B,B.1). While the TrkB immunoreactivity was intense in nerve fibers lining Schwann cells and satellite glia (Figure 4A.1), TrkC was clearly visible in the cytoplasm and nerve fibers of SGNs (Figure 4B.1). Consistent with previous results from Johnson et al. 2017 [21], p75^NTR^ resides in Schwann cells and satellite glia at GW09 (Figure 4C,C.1).

Colorimetric double immunostaining exposed TrkB and TrkC at GW12 in the spiral ganglion (Figure 4D), with TrkC being the most intense around SGNs and TrkB more inside the cytoplasm of the satellite glia/Schwann cells and neurons (Figure 4D, color-deconvoluted inlet). A clearly visible localization for both receptors is obvious at GW16 and all following stages of SGN development. TrkB staining is allocated to the satellite glia and Schwann cells (Figure 4F). TrkC staining is present more around the cytoplasmic membrane of SGNs (Figure 4F, inlet up). At later stages, the staining intensity for the Trk receptors increase and the distribution appears less distinct (Figure 4H). Interestingly, elevated levels of the receptor protein expression accompany the downregulation of *BDNF* in ISH and RNAseq data.

p75^NTR^ shows intense staining in Schwann cells that surround nerve fibers (Figure 4C,C.1), in satellite glia, and in putative fibrocytes around the spiral ganglion. At GW12, p75^NTR^ was strongly expressed in satellite glia (Figure 4E) and becomes more intense in Schwann cells at GW16–GW18 (Figure 4G,I). In this period, *p75^NTR^* transcripts are downregulated such as *NT-3* and *BDNF* in RNAseq analysis. p75^NTR^-positive cells around the spiral ganglion lose IR more and more by GW18 (Figure 4I).

### 2.4. Expression of BDNF, TrkB, TrkC, and p75^NTR^ in the Sensory Epithelium

#### 2.4.1. BDNF RNA Transcripts in the Sensory Epithelium

Next, we measured the *BDNF* transcripts in the sensory epithelium of the apical, middle, and basal cochlear duct in manually outlined ROIs. Quantification exposed *BDNF* to be highly upregulated at the early stages (GW09) and a gradual downregulation in the sensory epithelium until GW12–GW13. Around GW15 to GW16, the *BDNF* ISH signal increased with a pronounced expression in the apical turn (Figure 5A,B).

At the microscopic level, we confirm the previous results [21] and detect a high ISH signal at GW10 and GW11 in the GER as well as LER and surrounding tissue (Figure 5C).

At GW12, *BDNF* ISH is much weaker and restricted to the apical and basal portion of GER, most intensely in hair cells in the LER (Figure 5E). This expression pattern of *BDNF* remains at GW16 (Figure 5G–K) and is intensified in all hair cells and supporting cells, most intensely in Deiters phalangeal cells. The tonotopical maturation gradient visible at GW15/GW16 coincides with an apical-to-basal gradient in the intensity of the *BDNF* expression in GER as well as LER. In the more immature apical turn, the *BDNF* ISH staining intensity is the highest in the IHCs, phalangeal heads of the Deiters cells, OHCs, and the apical pole of GER cells (Figure 5G). The middle turn and basal turn of GW16 are characterized by a high *BDNF* ISH signal in hair cells and pillar cells as well as cells of the GER (Figure 5I,K). This ISH staining pattern remained constant until GW18. At GW18, we saw a reduced *BDNF* ISH signal with the localization in the apical portion of the hair cells and Kölliker’s organ (Figure 5M). For each developmental stage, we used sequential sections as the negative control ISH with a sense riboprobe (Figure 5D,F,H,J,L,N).

The tonotopical gradients of the *BDNF* expression differ between LER and GER in the early stages until GW12. The apical-to-basal gradient of the *BDNF* signal at GW15–GW16 coincides with a differentiation gradient of the organ of Corti. Hair cells, surrounding supporting cells, as well as supporting cells of Kölliker’s organ are the main source of *BDNF* transcripts.

#### 2.4.2. Receptor Immune Staining for the Sensory Epithelium

Congruent with a previous report by Johnson et al., 2017 [21], we could not find TrkB/C staining until GW11 within the hair cell domain, but we found a weak staining of TrkB in Kölliker’s organ (Figure 6A,B). Some weak IR for p75^NTR^ was present between GER/LER cells, indicating nerve fibers invade the cochlear duct (Figure 6C).

At GW12, the positive IR for TrkC in nerve fibers marks the innervation of the inner and outer hair cells. TrkB IR is present in GER cells and confirms the staining pattern in Johnson et al., 2017 [21] with a different cutting direction that led to a different interpretation of localization in that study (Figure 6D,F,H). This expression pattern is present in all the following investigated developmental stages. p75^NTR^ IR at GW12 marks the massive invasion of nerve fibers that innervate these sensory cells and even expose the fiber overshoot that we previously observed in earlier stages [13]. Supporting cells underneath the hair cells also show p75^NTR^ IR. Later stages enable a more exact localization and identify Deiters cells and pillar cells with a high p75^NTR^ IR. (Figure 6E,G,I).

## 3. Discussion

We examined the human cochlea from GW09–19 to cover the most important stages of hearing organ development. We evaluated the relative cochlear expression changes of the neurotrophic factors and receptors with RNAseq analysis and added ISH for *BDNF* and TrkB/TrkC/p75^NTR^ for correlation. The overlap between RNAseq and ISH data for BDNF demonstrates the reliability of these methods for detecting spatiotemporal expression patterns.

RNAseq data representing a snapshot of the transcriptome of the entire cochlea correlate in their profile of gene regulation better with the expression profile of spiral ganglion ISH quantification than sensory epithelium *BDNF* transcript quantification. This suggests that the spiral ganglion is one of the main sources of *BDNF* in our samples.

The phasic expression of *BDNF*, other neurotrophins, and their receptors suggest that this correlates with certain developmental steps. *BDNF* peaks in a first wave in the SG as well as sensory epithelium around GW10. This corresponds to a time point when hair cells differentiate, stereocilia emanate [28], nerve fibers reach hair cells in the basal turn [13], and afferent synapses begin to form.

At this early stage of fetal development, *BDNF* transcripts are not restricted to sensorineural cell types but also include mesenchymal cells around the primordial cochlear duct and cartilage cells. *BDNF* is known to promote bone formation and maturation [29] and may explain high expression levels around GW10/GW11, represented by the upregulation of *BDNF* expression at GW11 in RNAseq. The otic capsule develops from mesenchymal cells that differentiate into embryonic cartilage, and, later, into bone. The expression of p75^NTR^ in mesenchymal cells was described early [30] and it was later confirmed that the pluripotent mesenchymal stem cells express BDNF and NGF but not NT-3 and NT-4 [31]. Recently, BDNF was found to affect osteoblast differentiation through the TrkB receptor, and the JNK and p38 MAPK signal pathways [32], which explains the BDNF expression in these cell types during the formation of the otic capsule.

Sensorineural development around GW10 implies hair cells’ cell cycle exit [13] and differentiation in a basal-to-apical gradient, whereas the SGN differentiated two weeks earlier [13]. The SGN *BDNF* levels follow a distinct basal–apical *BDNF* gradient only at GW10 and matches the GER gradient. *BDNF* transcripts in Kölliker’s organ and some weak TrkB IR in the same site suggest the autocrine stimulation of this tissue by BDNF. Here, epithelial cells are still mitotically active as we showed previously [13], which suggests it to be in a primordial state at GW10, distinguished by its multilayered nuclei. Before the hair cells can receive acoustic stimulation, this transient epithelium produces ATP-mediated Ca^2+^-driven inward currents that change the morphology of this cochlear duct portion and cause spontaneous electric activity. These rhythmic transients are thought to coordinate spontaneous electrical activity with its neighboring IHC [33] and act as an important factor in establishing correct tonotopical wiring upstream the brain stem. The autocrine stimulation of Kölliker’s organ supporting cells by BDNF/TrkB could be one promoter for the maturation and initiation of this depolarization.

BDNF may also already be involved in the attraction of synaptophysin-positive nerve fibers in humans at GW8 [13], that even overshoots this epithelium and precedes hair cell differentiation. The expression of *BDNF* in immature hair cells and LER supporting cells without immunohistochemical proof of TrkB receptors suggest that BDNF also acts here mainly as a guidance cue to attract nerve fibers. SGNs are ahead in maturation and show the most intense *BDNF* transcripts in their satellite glia cells. Together with our results of TrkB IR in satellite glia and Schwann cells, BDNF may mainly act via their glia cells, whereas SGNs express TrkC that activates through NT-3. High NT-3 levels at GW11 in RNAseq data indicate support for this theory. The co-activation of BDNF and p75^NTR^ present in SG glia cells may trigger the differentiation and growth [34] of cochlear neuroglia and also, indirectly, SGNs.

Reports about the opposing tonotopical gradients of NT-3 and BDNF expression during development was summarized by Green S. et al., 2012 [35], and changes in BDNF expression that may cause a change in innervation patterns in rodents [27] highlight the importance of concerted spatiotemporal BDNF expression in sensory, epithelial, and neuronal cells [36]. However, a gradient of *BDNF* expression is also obvious in the human GER region and SGNs with the highest content in the base and the lowest in the apex at GW9–10. This contradicts the animal data with a reverse tonotopical gradient [14,37]. In later stages, at GW15/GW16, we quantified the highest *BDNF* expression in the more immature apex and less in the more advanced basal high frequency region. In the LER region containing the OHCs, we see the lowest content in the base but the highest in the middle turn. These data show that the description of NTF gradients need to be handled with caution. They may be different depending on the exact site of a cochlear duct and time point and among different species.

The huge drop of *BDNF* at GW12 coincides with big changes in the cochlea: the scala vestibuli and tympani begin to form, SGNs migrate to the central modiolus, and hair cells express SOX2 and MyoVIIa [12] in basal IHCs. The otic capsule and most mesothelial cells lose BDNF expression. This indicates some yet unknown maturation processes in the cartilage matrix that start to mineralize not before GW19 [38]. The staining pattern and tonotopical gradients of sensorineural components are very similar to GW10. SGNs send out peripheral axons and reach IHCs and OHCs [13], which show the highest *BDNF* expression in ISH in a base-to-apex gradient, but no Trk receptor IR. Kölliker’s organ still expresses BDNF and TrkB for a possible continuation of autocrine stimulation. *BDNF* transcripts further concentrate around mesothelial cells around the GER and satellite glia cells that also express TrkB.

The second peak of *BDNF* at GW15/GW16 is distinguished by mature scalae formation and the concentration of the *BDNF* expression to the SGNs/satellite glia, hair cells/Deiters cells, lateral wall, and supporting cells of Kölliker’s organ. Glia-like supporting cells surrounding hair cells are required for cell patterning, planar cell polarity, and synaptogenesis in the developing sensory epithelia, and supporting cells secrete multiple factors that act on hair cells and/or sensory neurons through reciprocal interactions to modulate synaptic connections [39]. *BDNF* persists in rodents even until adulthood around the IHC [39] and is important for sensorineural survival. BDNF is released by the hair cells and surrounding supporting cells and attracts fiber outgrowths [8,40,41]. In the sensory epithelium, we found the *BDNF*/TrkB co-expression at high levels only present in supporting cells of Kölliker’s organ, which speaks again to the presence of autocrine stimulation. Since p75^NTR^ is lacking in this epithelium, there is no indication for any apoptotic pathway triggered by *BDNF*. p75^NTR^ is present only in the nerve fibers that innervate IHCs and OHCs and both pillar cells. p75^NTR^ may be involved in the formation of Corti’s tunnel, and the co-expression of TrkB (satellite/Schwann cells)/TrkC (SGNs) and p75^NTR^ in growing nerve fibers enhances sprouting and elongation. Tonotopic gradients in the LER region exposes the highest *BDNF* content in the apex and the lowest in the base, opposing the gradient in GW8. This switch in tonotopic *BDNF* gradients coincide with the hair cell maturation steps. While the cell cycle exit is first detectable in the apical turn at GW7 [5], hair cells differentiate in the cochlear duct in a basal-to-apical gradient, in the basal portion of the cochlea starting at GW11/GW12 [12]. Once the hair cell innervation is completed around GW14 to GW15 [12], the phase of nerve fiber pruning starts [42]. Pruning is related to neurite refinement and the retraction of immature SGNs as well as neuronal apoptosis [43]. *BDNF* levels in the SGNs between GW14–GW18 are more complex than the clear gradient in GW10 and expose the middle turn neurons as the main *BDNF* producers. The hair cell innervation density is highest in the middle turn [44] so a higher SGN density and higher dynamic of innervation/pruning is likely the reason for this *BDNF* transcript distribution. The shift of the peak expression by two weeks in the apical turn likely represents the gradient of cochlear duct/SGN development. A lower innervation density in the basal turn but earlier maturation compared to the apex results in a more constant but rather low expression in the high-frequency region. Interestingly, an abrupt shutdown of *BDNF* levels appears at GW18 in all cochlear turns. This likely marks the completion of the *BDNF* action in cochlear development and results in a cochlea with an adult size. Some weak signal resides in the SGNs, hair cells, and supporting cells of Kölliker’s organ. Likewise, receptor levels rise for TrkB (Schwann and satellite cells), TrkC (SGNs), and p75^NTR^ (Schwann and satellite cells). This opposing regulation of the ligand–receptor combination conforms to the time course in RNAseq data for *NT-3-TrkC* and *BDNF-TrkB* and the course of *p75^NTR^* levels. Since *BDNF* transcripts are under the control of 11 different promoters and untranslated exons that are alternatively spliced and encoded for different levels of BDNF production, regulation is complex and not yet understood completely. A switch to a promoter with a more basal level of BDNF production to maintain some neurotrophic “survival” signaling could be one explanation. Alternatively, after the completion of the pruning phase and the establishment of tonotopicity around GW18 to GW19, nerve guidance and attraction cues are no longer necessary and neurotrophic production shuts down [6,8]. The *p75^NTR^* levels follow the same course, probably to match the neurotrophic receptor levels to ensure trophic signaling instead of apoptotic [34].

NGF via TrkA signaling is not thought to be relevant for inner ear development [45,46,47]. In any case, we were able to detect transcripts in RNAseq analysis also with an opposing regulation of ligand vs. receptor, like the other neurotrophins of this study. Thus far, we were never able to detect TrkA receptor proteins in our human samples, despite IR in the facial nerve. Due to the vicinity of this VII^th^ cranial nerve to the VIII^th^ vestibulo-chochlear nerve, we cannot exclude some facial nerve remnants in our cochlear preparation.

*NT-4* analyzed in RNAseq data presents a biphasic course that opposes all other neurotrophins and neurotrophic receptors. Although NT-4 was not found in the cochlea in earlier studies [48], newer sensitive methods like ours are able to detect much lower levels of transcripts, and mouse tissue may be different from human. The source of NT-4 expression in the human cochlea remains elusive, and, since neurotrophins act mainly as short-range ligands, we can only speculate about the site of production and signaling. BDNF and NT-4 lead to the differential endocytic sorting of TrkB receptors, resulting in rapid internalization, while the surface receptor was sustained longer with NT-4, which was capable of maintaining longer sustained downstream signaling activation [49]. This suggests NT-4 acts in a complementary way on the TrkB receptor to maintain a basal level of activation at developmental stages when BDNF transcripts are downregulated.

## 4. Conclusions

SGNs appear to be the main source of *BDNF* expression during human fetal development and peak at GW10 and once more at GW15–GW17, proofed by ISH and RNAseq. This coincides with periods of hair cell maturation and innervation. Not only SGNs, but, even more, satellite glia cells present an important source for BDNF expression. Tonotopical gradients of *BDNF* expression revert between GW8 and GW15 in the LER region and largely follow the maturation gradient in SGN/satellite cells and the GER region. *NT-3/TrkC* follows the same time course of expression levels as *BDNF/TrkB*. Due to the IR results, TrkB signaling directly affects the satellite glia, Schwann cells, and supporting cells of Kölliker’s organ, while TrkC signaling affects SGNs and pillar cells in the sensory epithelium in human development. The mission of neurotrophins likely includes nerve guidance, maturation, and trophic effects. The upregulation of *NT-4* may cover periods with a low *BDNF/NT-3* expression for sustained survival signaling along with fetal development (Summarized in Table 1). Future studies should explore the regulatory mechanisms underlying neurotrophic control to better understand the downstream action during human development and address new therapeutic strategies for sensorineural disorders.

## 5. Materials and Methods

### 5.1. Fetal Specimens and Ethical Approval

Human specimens (between the GW12 and GW19) were provided by the UCL London and Newcastle branches of the HDBR: Joint MRC/Wellcome Trust (grant # MR/R006237/1) Human Developmental Biology Resource (http://hdbr.org, accessed on 28 November 2024). Fetal and embryonic tissue was collected, with informed consent, and distributed to research projects under ethical approval 18/NE/0290 from the North East-Newcastle & North Tyneside 1 Research Ethics committee for HDBR Newcastle and 18/LO/022 from the Fulham Research Ethics Committee for HDBR UCL London. Specimens were certified by embryologists to exhibit no visible malformations and their embryological stages were differentiated by quantifying characteristics like crown–rump length, external and internal morphology, and the estimated gynecological age. All specimens were devoid of any external or internal congenital defects.

### 5.2. Tissue Preparation for Histology, Immunohistochemistry, and In Situ Hybridization on Paraffin Sections

Twenty-nine human fetuses (GW12 x3, GW13 x4, GW14 x4, GW15 x4, GW16 x4, GW17 x3, GW18 x3, and GW19 x4 as biological replicates) were used for in situ hybridization and immunohistochemistry. Tissue preparation for paraffin embedding, immunohistochemistry staining, and digital acquiring of human fetal specimens were described in detail in previous publication [21,50,51]. Immunochemistry and ISH sections of human fetus specimens (GW09 x1, GW10 x3, GW11 x1, and GW12 x2) used in previous publication of Johnson Chacko, 2017 [21] were re-imaged and re-analyzed.

### 5.3. Immunohistochemistry and Image Analysis

Immunohistochemistry was performed on a Leica Bond RX immunostainer (Leica Biosystems, Nußloch, Germany) applying standard procedure for colorimetric double staining. Then, 5 µm-thick FFPE human fetal inner ear sections were incubated with each primary antibody: p75^NTR^ (monoclonal, rabbit, 1:500, Abcam, Cambridge, UK, Cat. Nr. ab52987), TrkB (monoclonal, rabbit, 1:266, Cell Signalling, Leiden, The Netherlands, Cat. Nr. 4607), and TrkC (monoclonal, rabbit, 1:500, Cell Signalling, Cat. Nr. 3379) for 40 min at 37 °C and the Universal Secondary Antibody at 37 °C for 40 min (supplied in used detection kit), visualized with the detection systems (DAB) BOND Polymer Refine Detection (REF: DS9800, Leica Biosystems, Nußloch, Germany) and Bond Polymer refined Red Detection (REF: DS9390, Leica Biosystems, Nußloch, Germany). Stained sections were digitally examined using ZeissAxio Imager M2 microscope coupled to an Axiocam 512 color camera (Zeiss, Jena, Germany).

### 5.4. Riboprobe Synthesis for In Situ Hybridization

Human-BDNF-specific riboprobes were synthesized using the following primers: forward 5′-ATTTAGGTGACACTATAGAAGAGGGCTGACACTTTCGAACACA-3′; reverse: 5′-TAATACFACTCACTATAGGGAGACTTATGAATCGCCAGCCAAT-3′. The DNA product was 519 base pairs long and was synthesized using Go-Taq Green Master Mix (Promega, Madison, WI, USA) and cDNA-reverse-transcribed from mRNA isolated from human inner ear tissue. For the PCR reaction set up, the following is used: 40 cycles with denaturation at 95 °C 40 s, annealing at 60 °C 40 s, extension at 73 °C 40 s, and final synthesis at 73 °C 5 min.

For production of the antisense BDNF riboprobes, the T7 RNA polymerase promoter sequence (5′-TAATACGACTCACTATAGGGAGA-3′) was added to the forward primer, and for sense BDNF riboprobe, and the SP6 RNA polymerase promoter sequence (5′-ATTTAGGTGACACTATAGAAGAG-3′) was added to the reverse primer. PCR product was synthesized using the same conditions as above. Then, 100ng PCR product was Sanger-sequenced by Microsynth (Vienna, Austria) using T7 and SP6 primers. The identification and orientation using the promoters was controlled using NCBI Blast (NIH, Bethesda, MD, USA) nucleotide sequence alignment tool (Table 2). The antisense and sense orientation of the sequence following T7 and SP6 promoters was confirmed. For control purposes, T7-conjugated sense control riboprobe was used, which showed only a minimal background reaction in the cochlea and other inner ear tissue.

Antisense and sense riboprobes were synthesized and digoxigenin (DIG)-labelled using the T7 and SP6 in vitro transcription kit from Roche Life Sciences (Cat. No. 11 175 025 910, Roche, Mannheim, Germany), and 1 μg of template PCR products. The DIG labelling and the riboprobe concentration were determined using the DIG luminescent detection kit (Cat. Nr. 11 363 514 910, Roche) and CDP-star substrate (Roche) following the instructions of the manufacturer, Roche Life Sciences.

### 5.5. In Situ Hybridization and Image Analysis

In situ hybridization was performed on 5 µm-thick paraffin sections of human fetal inner ear samples between GW12 to GW19 to identify the expression level of *BDNF* in different developmental stages.

The ISH was performed on a Ventana Discovery Ultra immunostainer (Mannheim, Germany) using the Ribomap and Bluemap kit (REF: 760-120, Ventana Roche, Mannheim, Germany). The antigen retrieval was carried out with CC1 mild buffer and Protease 3 (15 min) (REF: 760-2020, Ventana Roche, Mannheim, Germany). The hybridization step was performed at 66 °C for 3 h with 200 ng/mL DIG-labelled specific riboprobe, 160 µg/mL sheared salmon sperm DNA (AM9680, Ambion, Vienna Austria), and 100 µL Ribohybe (manually prepared) on each slide. The sense and antisense signals were detected with an anti-DIG Fab fragment antibody (alkaline phosphatase coupled, incubated for 16 min) and by using the Bluemap Kit (Roche, Ventana) as instructed by Roche (incubation 1.5 h). Tetramizole (70 mg Tetramizole in 35 mL reaction buffer) was used for masking the endogenous alkaline phosphatase reaction. The counterstain was performed with Red Counterstain II (REF: 780-2218, Ventana, Roche, Mannheim, Germany). The antisense riboprobe represents the specific expression reaction, while the sense probe did not yield any reaction.

The in situ slides were digitally acquired at 40× magnification (Plan-Apochromat Air Zeiss, Jena, Germany) using a TissueFax Plus System coupled onto a Zeiss^®^ Axio Imager Z2 Microscope (TissueGnostics^®^, Vienna, Austria). The intensity of the BDNF signal was evaluated using the dedicated software HistoQuest^®^ 7.0 (TissueGnostics). Regions of interest (SGN, GER, LER, and Cochlea) were manually segmented. GER (inner hair cell, inner phalangeal cells, inner pillar cells, and Kölliger’s organ, and future interdental cells) and LER (outer hair cell, outer pillar cell, and cells of future outer sulcus) nomenclature was used until the tunnel of organ of Corti was formed (around GW15 to GW16). After GW15 to GW16, the regions were defined as “Corti with correlated region”. For statistical analyses, in situ BDNF “Sum Intensity Expression” was measured and exported to Excel 2016. Graphical analyses were carried out with Graph Pad Prism 10 (La Jolla, CA, USA).

### 5.6. RNA Extraction and Next-Generation Sequencing

Twenty-one human fetuses (GW12 x2, GW13 x2, GW14 x3, GW15 x4, GW16 x3, GW17 x2, GW18 x1, and GW19 x1 as biological replicates) were used for RNAseq analysis. For RNA extraction from inner ear samples, we used a combination of the method Ambion Trizol (Cat. No. 15596018, Invitrogen, Darmstadt, Germany) and RNeasy Micro Kit (Ref: 74004, Qiagen, Hilden, Germany). Generally, the extraction process included homogenization of the tissue, protease digestion, binding to solid substrate, washing, and elution. Purification of extracted RNA was performed by following the procedure protocol DNA&RNA precipitation manual from Genelink (Cat. No. 40-5135-05, Hawthorne, NY, USA) with ammonium acetate. RNA quantity was measured with a BioPhotometer plus (Eppendorf, Hamburg, Germany) and Qubit Fluorometric Quantification (ThermoFischer, Karlsruhe, Germany). RNA purity was defined with A260/280 and A260/230 absorbance ratios. The integrity of 28s and 18s rRNAs was determined using the Bioanalyzer 2100 from Agilent (Santa Clara, CA, USA).

The 3′ mRNA sequencing libraries were created using the Quant Seq 3′ mRNA-Seq Library Prep Kit (Lexogen, Vienna, Austria) according to manufactural instructions. Finally, RNA sequencing was performed on an ION Proton platform (Thermo Fisher, Karlsruhe, Germany), according to manufacturer’s instructions, yielding 7–8 million reads per sample. Raw RNAseq data were pre-processed using the https://nf-co.re/rnaseq (accessed on 28 November 2024) pipeline (done with version 3.9, newest version is 3.17.0) [52,53]. The raw files (fastqc) were pre-processed in the nf-fore pipeline with trim-galore for quality control and adapter trimming. Alignments with STAR [54] RSEM tools v1.3.3 [55] were used for indexing the reference genome human hg38 and for mapping RNA-seq reads to the genome. Afterwards, in the pipeline processing, the data were sorted and aligned with SAMTools 1.18. The generated count matrix was then imported into the Bioconductor R package DESeq2 4.2 to generate normalized gene expression matrix and differentially expressed genes with log2 fold change > 1 and adjusted *p*-value/FDR of <0.5 plotted as DE expression. Custom R scripts for generating plots were used for the analysis and visualization (all codes are available from the corresponding author upon reasonable request) of RNAseq expression.

## Figures and Tables

**Figure 1 ijms-25-13007-f001:**
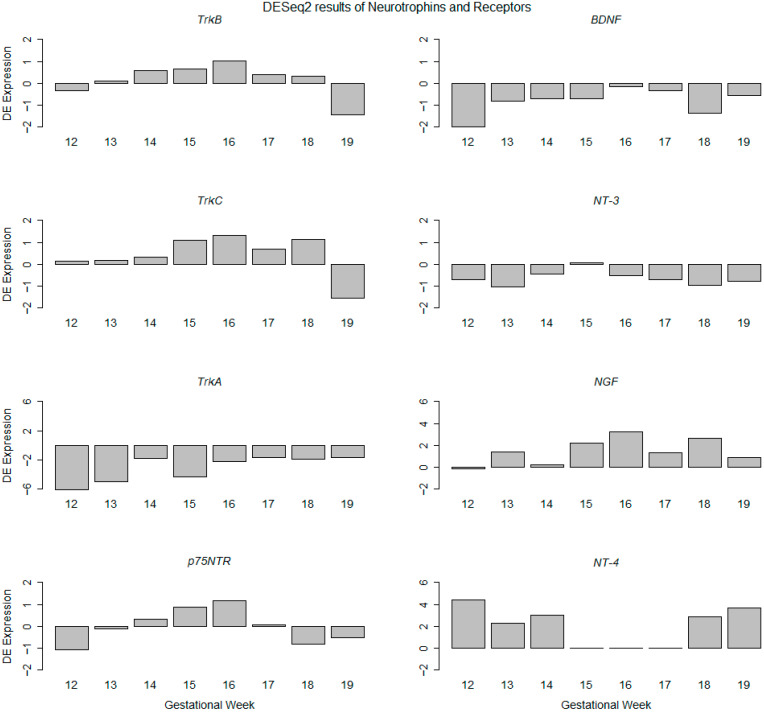
Differential expressed (DE) genes identified from the RNAseq data of neurotrophin receptors (TrkA, TrkB, TrkC, and p75^NTR^) and their ligands (BDNF, NT-3, NT-4, and NGF) genes between GW12 to GW19 relative to GW11. The RNAseq profile is represented as DE expression of differentially expressed genes (*y*-axis) and gestational week (*x*-axis). GW11 is taken as the experimental calibrator for RNAseq data. Compare also Appendix A for a heat-map-coded presentation of data.

**Figure 2 ijms-25-13007-f002:**
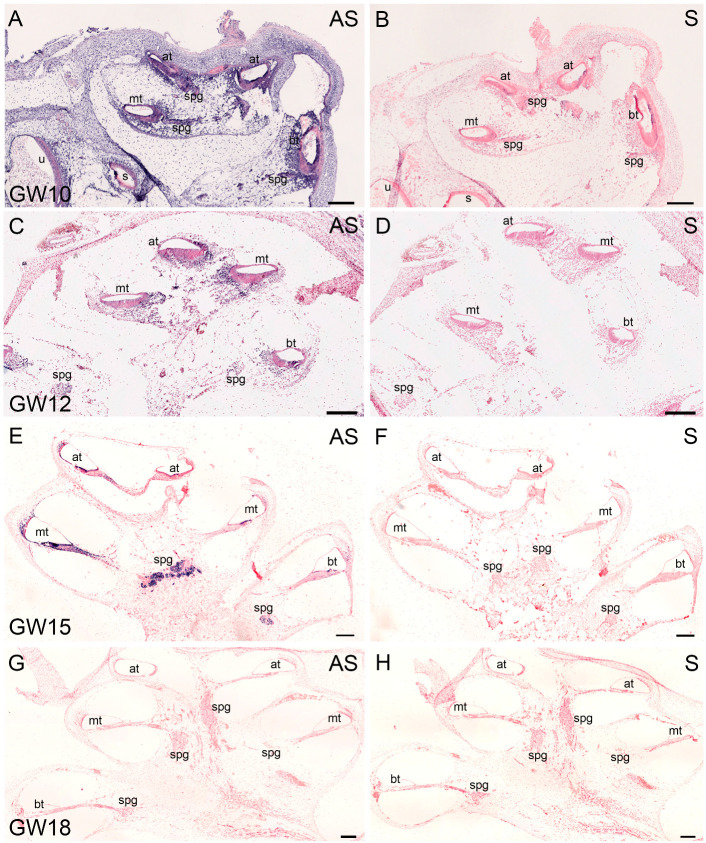
BDNF expression in sections of human fetal cochleae from GW10 to GW18. Purple signal represent positive detection. Counterstaining: nulcear fast red. (**A**) Expression of BDNF at GW10. Intense staining is present in the spiral ganglion and cochlear duct. (**C**) At GW12, overall expression of BDNF is diminished and concentrated around the cochlear duct and spiral ganglion. (**E**) At GW15, BDNF signal increases but is restricted to the spiral ganglion, and cochlear duct, as well as spiral ligament. (**G**) At GW18, BDNF ISH is downregulated. (**B**,**D**,**F**,**H**) BDNF-S: sense ISH is the negative control for antisense reaction. at: apical turn; mt: middle turn; bt: basal turn; spg: spiral ganglion; u: utricle; s: saccule; AS: antisense BDNF ISH, S: sense BDNF ISH; scale bar (**A**–**H**): 200 µm.

**Figure 3 ijms-25-13007-f003:**
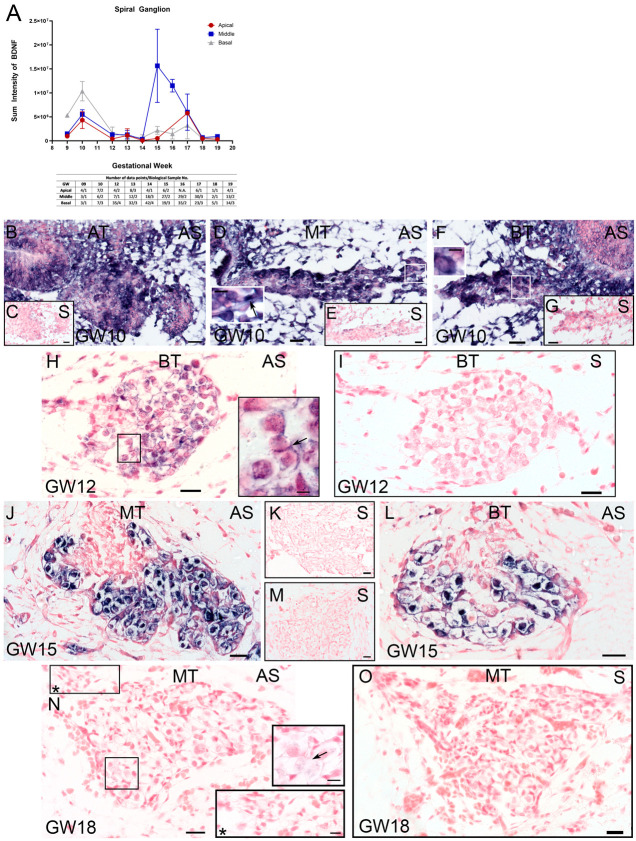
BDNF expression in the human fetal spiral ganglion from GW09 to GW19. (**A**) BDNF sum intensity in tonotopical areas (apical, middle, and basal) plotted as line graph. *x*-axis depicts the developmental stages and *y*-axis showed the measured sum intensity values. The table underneath the graph presents the number of ROIs measured (number of data points) from different biological samples (biological sample number). (**B**,**D**,**F**) BDNF expression at GW10 in SGNs and glial cells (**F Inlet**). (**C**,**E**,**G**,**I**) BDNF- sense ISH negative control. All sections counterstained with nuclear fast red. (**H**) BDNF ISH signal at GW12 in SGNs and satellite glial cells (**H inlet, arrows**). (**J**) High intensity of the BDNF signal in the spiral ganglion of the middle turn. (**L**) Basal turn at GW15 with lower BDNF signal intensity. (**N**) BDNF antisense signal at GW18 with low levels in the spiral ganglion (inlet, arrow) (inlet with asterisk). (**K**,**M**,**O**) BDNF- sense ISH negative control. AT: apical turn; MT: middle turn; BT: basal turn; AS: antisense BDNF ISH, S: sense BDNF ISH; scale bar (**B**,**D**,**F**,**H**,**I**,**J**,**L**,**N**,**O**): 20 µm; scale bar (**C**,**E**,**G**,**K**,**M**): 5 µm.

**Figure 4 ijms-25-13007-f004:**
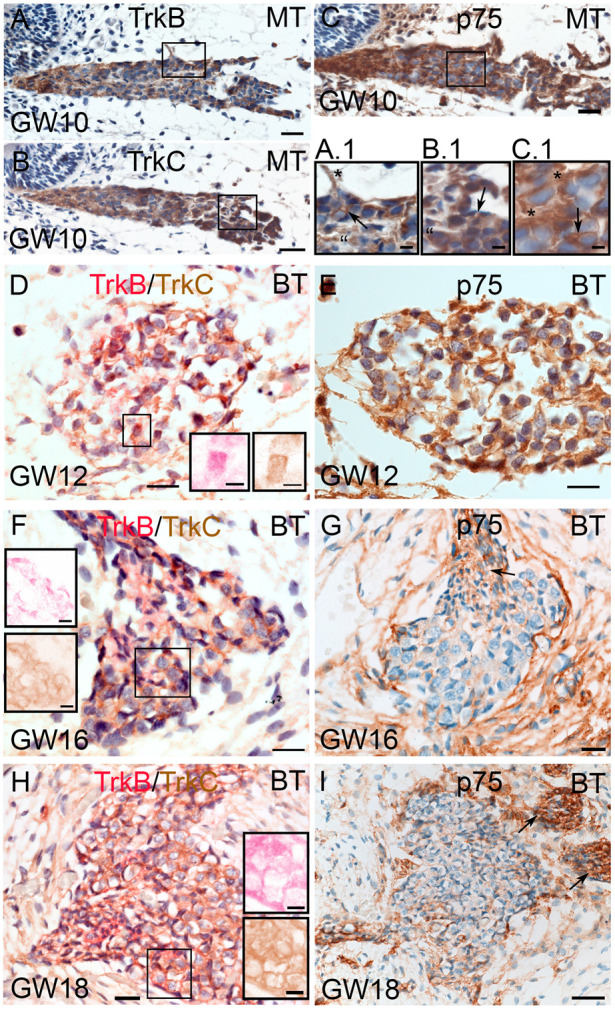
TrkB (red), TrkC (brown), and p75^NTR^ (brown) immunostaining in the spiral ganglion between GW10 to GW18. Cell nuclei are counterstained with hematoxylin. (**A**) GW10 TrkB immune labelling in the spiral ganglion. (**A.1**) High magnification of TrkB staining in nerve fibers (asterisk), Schwann cells (arrow), and satellite glial cells (quote mark). (**B**) TrkC in the spiral ganglion at GW10. (**B.1**) High magnification of TrkC in the cytoplasm (arrow and quote mark) of SGNs. (**C**) p75^NTR^ in the spiral ganglion at GW10. (**C.1**) Higher magnification of p75^NTR^ and IR in satellite glial cells (asterisks) and Schwann cells (arrow). (**D**) TrkB and TrkC IHC double staining at GW12. (**D Inlets**): Higher magnified view of framed area in (**D**) after color deconvolution (pink: TrkB and brown: TrkC). (**E**) p75^NTR^ in the spiral ganglion at GW12. (**F**) GW16 depicts the distinct localization of both receptors. (**F Inlet up**): Color deconvolution of TrkB (pink) staining in the glial and Schwann cells. (**F Inlet down**): TrkC (brown) color-deconvoluted image resides in TrkC around the SGNs. (**G**) GW16 p75^NTR^ immunostaining is present in Schwann cells (arrow) and SGN-surrounding fibrocytes. (**H**) GW18 TrkB and TrkC immune labelling intensifies. (**I**): GW18 p75^NTR^ IR is present in Schwann cells of nerve fiber bundles (arrows) and diminishes in surrounding fibrocytes. MT: middle turn; BT: basal turn; scale bar (**A**–**H**): 20 µm (**I**): 50 µm (**Inlets**) 5 µm.

**Figure 5 ijms-25-13007-f005:**
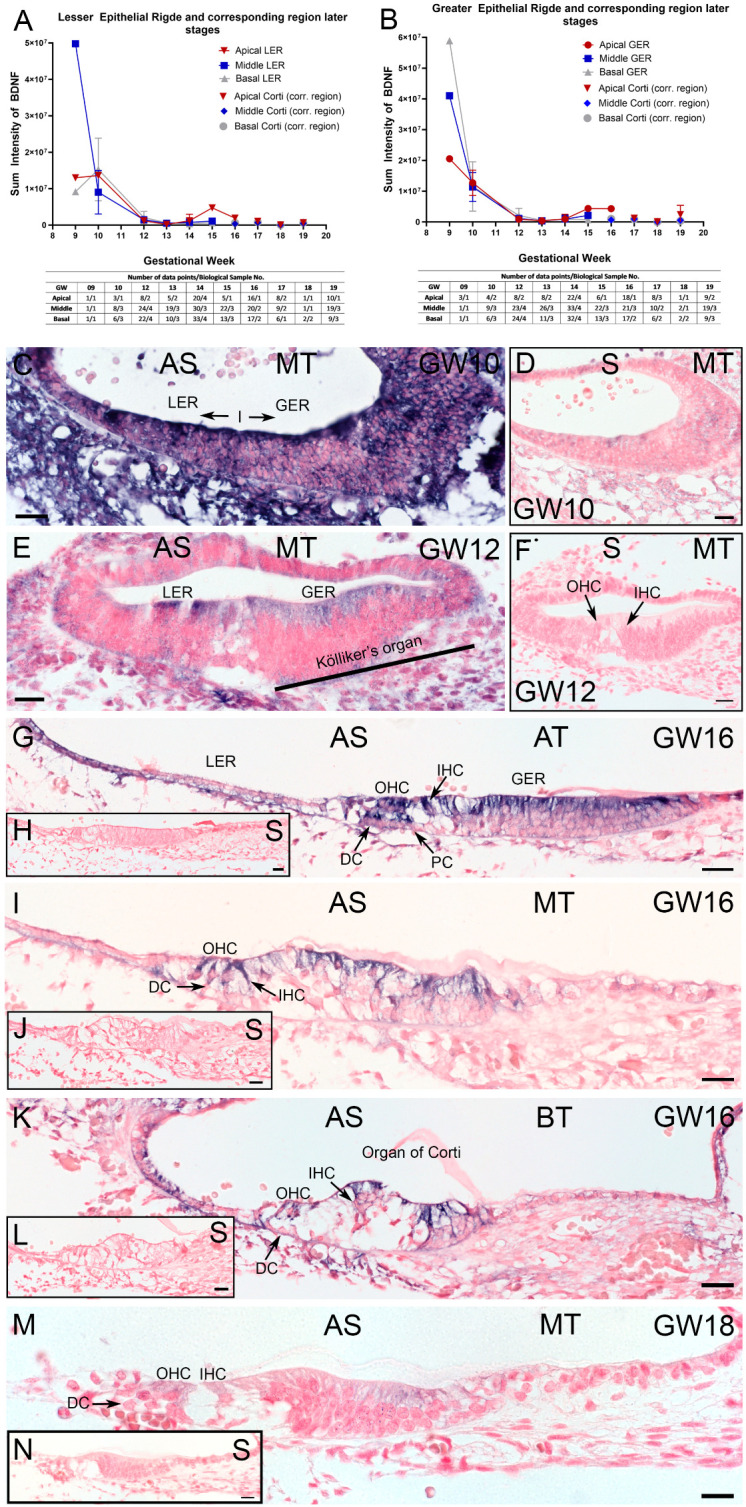
BDNF expression in the sensory epithelium between developmental stages GW09 to GW19. (**A**,**B**) BDNF sum intensity in distinct tonotopical areas (apical, middle, and basal) plotted as line graph for greater epithelial ridge (GER), lesser epithelial ridge (LER), and, at later stages, organ of Corti with corresponding region to GER/LER. BDNF staining gradually decreased in intensity until GW12, followed by a smaller second peak at GW15. Staining is pronounced in the apical turn and fades out until GW19. The tonotopical BDNF gradient in the GER reverts from GW9/10 to GW15/GW16. *x*-axis depicts the developmental stages and *y*-axis shows the measured sum intensity values. Table below under graphs showed measured ROIs (number of data points) from different biological samples (biological sample number). (**C**) At GW10, intense staining is present in the cochlear duct and surrounding tissue. (**D**) Sense control ISH section for GW10. (**E**) ISH signal dropped at GW12 and is most intense in hair cells. (**F**) Sense control ISH section for GW12. (**G**) BDNF increased at GW16, in the apical turn in the GER and LER including hair cells, PCs, and Deiters cells. (**I**) At GW16, middle turn BDNF was present in the GER, OHCs, PCs, IHCs, and supporting cells of Kölliker’s organ. (**K**) In the basal turn of GW16, intense BDNF signal is present in hair cells and Hensen cells. (**M**) With GW18, low BDNF ISH signal is confined to OHC and IHC and Kölliker’s organ. (**H**,**J**,**L**,**N**) BDNF-sense ISH as negative control for antisense reaction with nuclear fast red counterstaining. AT: apical turn; MT: middle turn; BT: basal turn; IHC: inner hair cell; OHC: outer hair cell, PC: pillar cell, DC: Deiters cell, GER: greater epithelial ridge, LER: lesser epithelial ridge, AS: antisense BDNF ISH, S: sense BDNF ISH; scale bar (**C**–**N**): scale bars 20 µm.

**Figure 6 ijms-25-13007-f006:**
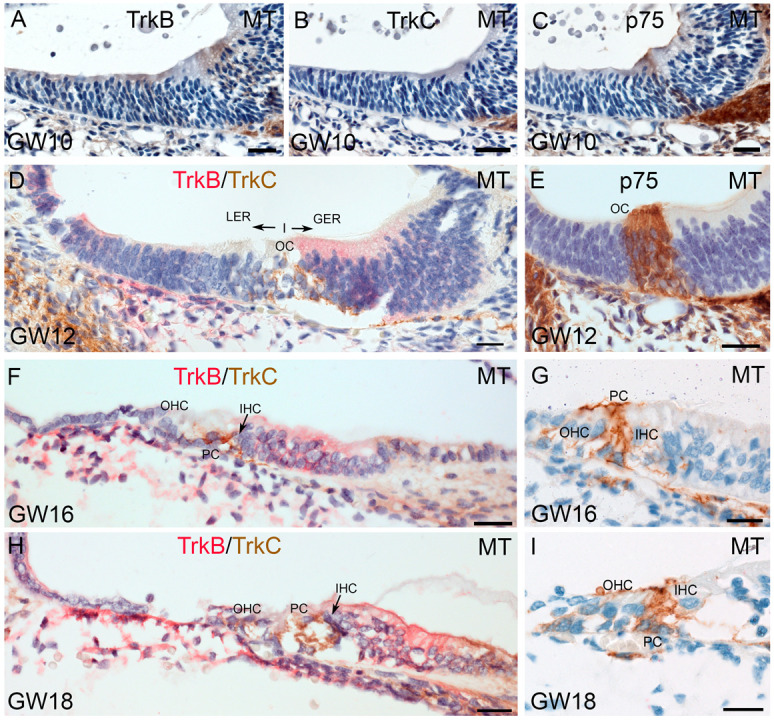
TrkB, TrkC, and p75^NTR^ immunostaining in the sensory epithelium between GW10 to GW18 of human fetal inner ears. (**A**,**B**) Weak immunoreactivity for TrkB was visible in Kölliker’s organ but is absent for TrkC in the sensory epithelium at GW10. (**C**) At GW10, the p75^NTR^ immunostaining presents in pillar cells and nerve fibers that innervate the future organ of Corti. (**D**,**F**,**H**) At GW12, TrkB staining was present in supporting cells of the GER region. TrkC staining was clearly visible in the putative afferent nerve fibers that innervate the OC. (**E**,**G**,**I**) p75^NTR^ immunoreactivity was present in nerve fibers and PCs. MT: middle turn; OC: organ of Corti; IHC: inner hair cell; OHC: outer hair cell, PC: pillar cell, GER: greater epithelial ridge; LER: lesser epithelial ridge; scale bar (**A**–**I**): 20 µm.

**Table 1 ijms-25-13007-t001:** Summary of expression timeline of BDNF, TrkB, TrkC, and p75NTR between GW09 to 19 in human inner ear. HC: hair cell, SC: supporting cell, PC: pillar cell, OC: organ of Corti; NF: nerve fiber; SPG: spiral ganglion; GER: greater epithelial ridge; SE: sensory epithelium; FMR: future modiolus region; ISH: in situ hybridization, SGC: satellite glial cell, ItR: intracellular region, LER: lesser epithelial ridge; DC: Deiters cells, HeC: Hensen cells.

Marker	Gestational Week										
	Summary of Johnson et al., 2017 [21]										
	09	10	11	12	13	14	15	16	17	18	19
BDNF	High concentration in early weeks, decreasing until GW12 SPG: in SGC and Schwann cells SE: intense in GER and LER and surrounding tissue			Restricted to SE and SPG SPG: higher concentration in SGC around SPG and in ItR of SGCs SE: weaker signal, restricted to apical and basal portion of GER, and intense in HC of LER			Increased expression at MT with GW15/16 and at GW17 in AT and decreased until GW18/19 SPG: faded in NF and Schwann cells SE: same pattern as in earlier weeks, at GW16, intensive in HC, supporting cells, DC phalangeal, and PC, decreased until GW18/19 with localization at apical portion of HCs and Kölliker’s organ				
TrkB/TrkC	SPG: TrkB intense in NF lining Schwann cells and SGC, TrkC in cytoplasm and NF of SPG SE: no visible staining			SPG: TrkC in NF of innervated OC and TrkB inside the SPG cytoplasm SE: TrkC in NF which innervate OC and TrkB in GER cells			SPG: local dissociation, TrkC visible around SPG, and TrkB present in SGC and Schwann cells SE: same staining of both Trks as in earlier weeks				
p75^NTR^	SPG: resides in Schwann cells and SGC SE: weak ItR between GER/LER region			SPG: intense in SG, Schwann cells, and putative fibrocytes around SPG SE: intensive in N; in later stages, also present in supporting cells underneath HC, DC, and PC							
RNAseq: NT-3, NT-4 NGF, (TrkA)	Data not available			TrkA: regarded as not important for cochlear development at this stage NT-3: upregulated at GW15 NT-4: upregulated in earlier (until 12) and later (GW18/19) stages NGF: upregulated around GW15/16							
Developmental Steps [6,8,12,41,43]	Development/ elongation to coiled structure with apical–middle–basal turn SPG lies adjacent to the GER region Undifferentiated SE	Start of SPG projection to central/FMR and NF innervation to OC Start of HC development		Neurite outgrowth and extension HC innervation Finishing of SPG projection to modiolus/central region	Neurite outgrowth and extension HC innervation		OC development finished HC innervation finished Start of pruning	Neurite refinement and retraction (Pruning)			Onset of hearing

**Table 2 ijms-25-13007-t002:** Sense and antisense sequences of BDNF probe for in situ hybridization.

*BDNF Sense Sequence*
Query	1	GAGGACCAGAAAGTTCGGCCCAATGAAGAAAACAATAAGGACGCAGACTTGTACACGTCC	60
||||||||||||||||||||||||||||||||||||||||||||||||||||||||||||
Sbjct	857	GAGGACCAGAAAGTTCGGCCCAATGAAGAAAACAATAAGGACGCAGACTTGTACACGTCC	916

Query	61	AGGGTGATGCTCAGTAGTCAAGTGCCTTTGGAGCCTCCTCTTCTCTTTCTGCTGGAGGAA	120
||||||||||||||||||||||||||||||||||||||||||||||||||||||||||||
Sbjct	917	AGGGTGATGCTCAGTAGTCAAGTGCCTTTGGAGCCTCCTCTTCTCTTTCTGCTGGAGGAA	976

Query	121	TACAAAAATTACCTAGATGCTGCAAACATGTCCATGAGGGTCCGGCGCCACTCTGACCCT	180
||||||||||||||||||||||||||||||||||||||||||||||||||||||||||||
Sbjct	977	TACAAAAATTACCTAGATGCTGCAAACATGTCCATGAGGGTCCGGCGCCACTCTGACCCT	1036

Query	181	GCCCGCCGAGGGGAGCTGAGCGTGTGTGACAGTATTAGTGAGTGGGTAACGGCGGCAGAC	240
||||||||||||||||||||||||||||||||||||||||||||||||||||||||||||
Sbjct	1037	GCCCGCCGAGGGGAGCTGAGCGTGTGTGACAGTATTAGTGAGTGGGTAACGGCGGCAGAC	1096

Query	241	AAAAAGACTGCAGTGGACATGTCGGGCGGGACGGTCACAGTCCTTGAAAAGGTCCCTGTA	300
||||||||||||||||||||||||||||||||||||||||||||||||||||||||||||
Sbjct	1097	AAAAAGACTGCAGTGGACATGTCGGGCGGGACGGTCACAGTCCTTGAAAAGGTCCCTGTA	1156

Query	301	TCAAAAGGCCAACTGAAGCAATACTTCTACGAGACCAAGTGCAATCCCATGGGTTACACA	360
||||||||||||||||||||||||||||||||||||||||||||||||||||||||||||
Sbjct	1157	TCAAAAGGCCAACTGAAGCAATACTTCTACGAGACCAAGTGCAATCCCATGGGTTACACA	1216

Query	361	AAAGAAGGCTGCAGGGGCATAGACAAAAGGCATTGGAACTCCCAGTGCCGAACTACCCAG	420
||||||||||||||||||||||||||||||||||||||||||||||||||||||||||||
Sbjct	1217	AAAGAAGGCTGCAGGGGCATAGACAAAAGGCATTGGAACTCCCAGTGCCGAACTACCCAG	1276

Query	421	TCGTACGTGCGGGCCCTTACCATGGATAGCAAAAAGAGAATTGGCTG	467	
|||||||||||||||||||||||||||||||||||||||||||||||
Sbjct	1277	TCGTACGTGCGGGCCCTTACCATGGATAGCAAAAAGAGAATTGGCTG	1323	

*BDNF Antisense Sequence*
Query	1	CCATGGTAAGGGCCCGCACGTACGACTGGGTAGTTCGGCACTGGGAGTTCCAATGCCTTT	60
||||||||||||||||||||||||||||||||||||||||||||||||||||||||||||
Sbjct	1301	CCATGGTAAGGGCCCGCACGTACGACTGGGTAGTTCGGCACTGGGAGTTCCAATGCCTTT	1242

Query	61	TGTCTATGCCCCTGCAGCCTTCTTTTGTGTAACCCATGGGATTGCACTTGGTCTCGTAGA	120
||||||||||||||||||||||||||||||||||||||||||||||||||||||||||||
Sbjct	1241	TGTCTATGCCCCTGCAGCCTTCTTTTGTGTAACCCATGGGATTGCACTTGGTCTCGTAGA	1182

Query	121	AGTATTGCTTCAGTTGGCCTTTTGATACAGGGACCTTTTCAAGGACTGTGACCGTCCCGC	180
||||||||||||||||||||||||||||||||||||||||||||||||||||||||||||
Sbjct	1181	AGTATTGCTTCAGTTGGCCTTTTGATACAGGGACCTTTTCAAGGACTGTGACCGTCCCGC	1122

Query	181	CCGACATGTCCACTGCAGTCTTTTTGTCTGCCGCCGTTACCCACTCACTAATACTGTCAC	240
||||||||||||||||||||||||||||||||||||||||||||||||||||||||||||
Sbjct	1121	CCGACATGTCCACTGCAGTCTTTTTGTCTGCCGCCGTTACCCACTCACTAATACTGTCAC	1062

Query	241	ACACGCTCAGCTCCCCTCGGCGGGCAGGGTCAGAGTGGCGCCGGACCCTCATGGACATGT	300
||||||||||||||||||||||||||||||||||||||||||||||||||||||||||||
Sbjct	1061	ACACGCTCAGCTCCCCTCGGCGGGCAGGGTCAGAGTGGCGCCGGACCCTCATGGACATGT	1002

Query	301	TTGCAGCATCTAGGTAATTTTTGTATTCCTCCAGCAGAAAGAGAAGAGGAGGCTCCAAAG	360
||||||||||||||||||||||||||||||||||||||||||||||||||||||||||||
Sbjct	1001	TTGCAGCATCTAGGTAATTTTTGTATTCCTCCAGCAGAAAGAGAAGAGGAGGCTCCAAAG	942

Query	361	GCACTTGACTACTGAGCATCACCCTGGACGTGTACAAGTCTGCGTCCTTATTGTTTTCTT	420
||||||||||||||||||||||||||||||||||||||||||||||||||||||||||||
Sbjct	941	GCACTTGACTACTGAGCATCACCCTGGACGTGTACAAGTCTGCGTCCTTATTGTTTTCTT	882

Query	421	CATTGGGCCGAACTTTCTGGTCCTCATCCAACAGCTCTTCTATCATGTGTTCGAA	475	
||||||||||||||||||||||||||||||||||||||||||||| |||||||||
Sbjct	881	CATTGGGCCGAACTTTCTGGTCCTCATCCAACAGCTCTTCTATCACGTGTTCGAA	827	

## Data Availability

Immunohistochemistry and ISH data are available upon reasonable request. Gene expression data will be uploaded to the EBI data repository.

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
