# Peer review of "Expression of Neurotrophins and Its Receptors During Fetal Development in the Human Cochlea"

_ijms, 2024, doi:10.3390/ijms252313007_

Round 1
Reviewer 1 Report
Comments and Suggestions for Authors
The manuscript explores neurotrophin expression and receptor localization, particularly of BDNF, in the human fetal cochlea during key developmental stages. This research is precious for understanding neurotrophic influences on inner ear maturation and offers potential insights into therapeutic approaches for hearing restoration. Below are five critical areas for improvement, alongside suggestions and specific grammatical corrections.
1) The RNA sequencing (RNAseq) methods lack details about data normalization techniques, criteria for defining significant expression changes, and quality control measures. While the authors note RNAseq was performed, the lack of specifics hinders reproducibility. Adding information on threshold values for differential expression and normalization methods, along with a summary of data quality control, would improve clarity and allow for methodological validation by other researchers.
2) The manuscript uses specialized terminology (e.g., “GER,” “LER,” “SGN”) without sufficient introductory explanations. Terms like these may limit the accessibility of the paper to a wider scientific audience. Providing brief definitions when these terms are first introduced and adding a glossary of abbreviations would make the paper more approachable for non-specialists and enhance readability.
3) Some figures lack scale bars, and tables do not consistently include units of measurement or clear explanations. For instance, figures illustrating RNAseq data need explicit legends detailing what each axis represents. Adding this information directly within figure legends would enhance understanding, prevent misinterpretation, and improve overall data presentation.
Comments on the Quality of English LanguageThe English could be improved to more clearly express the research.
Author Response
The respones to the reviewer comments are uploaded in the word document.

Reviewer 2 Report
Comments and Suggestions for Authors
The manuscript ijms-3305626 evaluates cochlear expression of neurotrophic factors and receptors using RNA-seq analysis and in situ hybridization (ISH). The results are valuable as they contribute to a better understanding of neurotrophic action in human inner ear development and may inform neurotrophic therapies for the treatment of hearing loss. Overall, the manuscript is well written, but several points require revision to improve the clarity and interpretation of the results.
General Comments
1. In the line 112 the authors refer to expression levels as "high" or in the line 117 as ‘increses” but present these as log fold changes. It would be clearer to explicitly describe the results in terms of fold changes or terms such as "upregulated" and "downregulated". In addition, the authors should consider presenting these results using a heatmap, which would provide a more intuitive and visually accessible representation of the data.
2. The caption of Figure 1 should indicate that these are differentially expressed (DE) genes identified from the RNA-seq data.
3. The labels in the ISH images are too large and make the figures look overloaded. Consider reducing the font size within the figures to improve clarity and readability.
4. The word "Citation" on line 201 should be removed.
Methods.
1. The manuscript does not specify how the quality of the sequencing data was assessed. For example, Trinity recommends using additional parameters to assess the quality of the transcriptome assembly. Consider referring to this resource for guidance: https://github.com/trinityrnaseq/trinityrnaseq/wiki/Transcriptome-Assembly-Quality-Assessment).
2. The RNA-seq data should be deposited in a public repository such as NCBI or EBI to comply with data sharing standards. This is not mentioned in the Data Availability section.
3. On line 574, the authors state "Gene expression was plotted as normalized FPKM reads". However, none of the plots show normalized FPKM values. Instead, the results show relative expression as log fold changes. This discrepancy should be corrected. The phrase in the Methods should be revised to accurately reflect the data presented in the manuscript.
Author Response
The respones to the reviewer comments are uploaded in the Word document.

Round 2
Reviewer 1 Report
Comments and Suggestions for Authors
Accept in present form